# Structural Comparison of Diverse HIV-1 Subtypes using Molecular Modelling and Docking Analyses of Integrase Inhibitors

**DOI:** 10.3390/v12090936

**Published:** 2020-08-26

**Authors:** Darren Isaacs, Sello Given Mikasi, Adetayo Emmanuel Obasa, George Mondinde Ikomey, Sergey Shityakov, Ruben Cloete, Graeme Brendon Jacobs

**Affiliations:** 1South African Medical Research Council Bioinformatics Unit, South African National Bioinformatics Institute, University of the Western Cape, Cape Town 8000, South Africa; 3433660@myuwc.ac.za; 2Division of Medical Virology, Department of Pathology, Faculty of Medicine and Health Sciences, Stellenbosch University, Francie van Zijl Avenue, P.O. Box 241, Cape Town 8000, South Africa; mikasi@sun.ac.za (S.G.M.); obasa@sun.ac.za (A.E.O.); 3Centre for the Study and Control of Communicable Diseases (CSCCD), University of Yaoundé 1, Yaoundé P.O. Box 8445, Cameroon; mondinde@yahoo.com; 4Department of Psychiatry & Mind-Body Interface Laboratory (MBI-Lab), China Medical University Hospital, Taichung 404, Taiwan; shityakoff@hotmail.com; 5Department of Bioinformatics, University of Würzburg, Würzburg 97074, Germany

**Keywords:** integrase, naturally occurring polymorphisms, HIV-1, molecular modelling, molecular docking, diversity

## Abstract

The process of viral integration into the host genome is an essential step of the HIV-1 life cycle. The viral integrase (IN) enzyme catalyzes integration. IN is an ideal therapeutic enzyme targeted by several drugs; raltegravir (RAL), elvitegravir (EVG), dolutegravir (DTG), and bictegravir (BIC) having been approved by the USA Food and Drug Administration (FDA). Due to high HIV-1 diversity, it is not well understood how specific naturally occurring polymorphisms (NOPs) in IN may affect the structure/function and binding affinity of integrase strand transfer inhibitors (INSTIs). We applied computational methods of molecular modelling and docking to analyze the effect of NOPs on the full-length IN structure and INSTI binding. We identified 13 NOPs within the Cameroonian-derived CRF02_AG IN sequences and further identified 17 NOPs within HIV-1C South African sequences. The NOPs in the IN structures did not show any differences in INSTI binding affinity. However, linear regression analysis revealed a positive correlation between the Ki and EC50 values for DTG and BIC as strong inhibitors of HIV-1 IN subtypes. All INSTIs are clinically effective against diverse HIV-1 strains from INSTI treatment-naïve populations. This study supports the use of second-generation INSTIs such as DTG and BIC as part of first-line combination antiretroviral therapy (cART) regimens, due to a stronger genetic barrier to the emergence of drug resistance.

## 1. Introduction

The HIV/AIDS pandemic continues to be a significant problem worldwide [1]. The viral integration process, which is the insertion of viral DNA into host genomic DNA, is an indispensable step of the retroviral life cycle and is catalyzed by the viral integrase (IN) enzyme [2]. Integration is achieved via two distinct sequential catalytic activities, 3′ processing and strand transfer. IN first processes viral DNA by excising a dinucleotide at the 3′ end, exposing hydroxyl ends. IN then catalyzes the introduction of the prepared DNA into genomic DNA by facilitating a nucleophilic attack upon genomic DNA [3,4]. The same active site in IN, which contains a retroviral highly conserved DDE motif and magnesium ions, performs both activities [5,6]. HIV-1 IN is a 32 kDa protein that functions as a tetramer or multimer [3,4]. A monomer consists of three distinct domains; the N-terminal domain (NTD) comprising residues 1–46, the catalytic core domain (CCD) comprising residues 56–186 within which the active site DDE motif (aspartate (D64), aspartate (D116), and glutamate (E152) is present, and the C-terminal domain (CTD) comprising residues 195–288 [7,8]. Several integrase strand transfer inhibitors (INSTIs) have been developed to target HIV-1 IN to prevent viral integration into the host genome. The four INSTIs available thus far include raltegravir (RAL) and elvitegravir (EVG) that are considered as first-generation inhibitors, while dolutegravir (DTG) and bictegravir (BIC), along with the late-phase clinically trialed cabotegravir (CBT), are classified as second-generation INSTIs [9]. At present, first-line combination antiretroviral therapy (cART) regimens for HIV-1 are expected to include the INSTI DTG according to World Health Organization (WHO) recommendations [10], as it has been shown to possess a higher genetic barrier to drug resistance development as compared with RAL and EVG [11]. HIV-1 is a genetically highly diverse virus, forming different subtypes, recombinant and region-specific variants [12]. Development of INSTIs, as with most pharmaceutical agents, was primarily conducted by companies in First World nations, where subtype B is the most predominant variant or subtype [13,14]. It remains unclear what effects naturally occurring polymorphisms (NOPs) may have upon the IN structure and INSTI susceptibility. This lack of data poses a challenge in concluding the effects of NOPs on the binding of INSTIs to HIV-1 IN subtypes [15,16,17,18,19].

In this study, computational methods, which include molecular modelling and docking, were used to determine if NOPs affect INSTI binding to HIV-1 IN subtype C (HIV-1C) and a circulating recombinant form of HIV-1 IN CRF_02_AG. The recently resolved cryogenic electron microscopy full-length HIV-1 subtype B IN structure allowed us to build accurate and complete tetrameric three-dimensional structures of HIV-1C IN and of HIV-1 IN CRF_02_AG. The value of having accurate protein models allows us to infer the exact mode of interactions formed between active site residues of HIV-1 subtypes and drug atoms. HIV-1 subtype C derived from a South African cohort was chosen as one of our IN models, as it represents the most prevalent subtype both globally and for sub-Saharan Africa in particular [14,19,20]. Our focus on a Cameroonian cohort was spurred on by the previously reported HIV-1 diversity present in Cameroon with all known subtypes/variants found within Cameroon [21,22]. Furthermore, the full-length Cryo-EM HIV-1 IN structure, which was used as the template in our molecular modelling, served additionally as a subtype B IN model in our study, the predominant strain in developed nations.

## 2. Materials and Methods

### 2.1. Ethics Statement

The study used sequences from two African settings: South Africa and Cameroon.

Ethical permission for this study was obtained from the Health Research Ethics Committee of Stellenbosch University (N14/10/130—approved on 13 August 2019 and N15/08/071—approved on 26 March 2019). Ethics protocols are revised and renewed each year. The study was conducted according to the ethical guidelines and principles of the international Declaration of Helsinki 2013, South African Guidelines for Good Clinical Practice, and the Medical Research Council (MRC) Ethical Guidelines for Research. A waiver of consent was awarded to conduct sequence analyses.

### 2.2. Study Design

HIV-1-positive plasma samples were obtained from the Centre for the Study and Control of Communicable Diseases (CSCCD), University of Yaoundé I, Cameroon (*n* = 37), and South Africa samples (*n* = 91) were requested, with permission, through the National Health Laboratory Services (NHLS) within the Division of Medical Virology, Stellenbosch University, South Africa. Samples were collected between March 2017 and February 2018 [23]. We excluded patient samples with no previous cART history and patients receiving first-line cART. Patients had their samples sent to the NHLS for HIV-1 genotypic resistance testing. Treatment failure is defined according to the South Africa adult antiretroviral guidelines by a confirmed viral load of >1000 copies/mL on two measurements taken two to three months apart.

### 2.3. Nucleic Acid Extraction

HIV-1 RNA extraction was performed using the QIAamp Viral RNA Mini Extraction Kit’s Spin protocol, according to the manufacturer’s instructions (Qiagen, Hilden, Germany). Briefly, 140 µL of plasma was used as a starting volume. A larger starting volume of 280 µL of plasma was used for samples with very low viral titers. Viral RNA was stored at −80°C until use.

### 2.4. PCR Amplification and Sequencing

The synthesis of complementary DNA (cDNA) and first-round PCR amplification were performed using the Invitrogen SuperScript^®^ III Reverse Transcriptase (RT) reagents (Invitrogen, Karlsruhe, Germany), as per the manufacturer’s instructions. In-house amplification of the IN region (867 bp, positions 4230–5096, HXB2 strain) by nested RT-PCR was performed as previously described by our laboratory [24,25]. Purified amplicons were sequenced on both strands with conventional Sanger DNA sequencing, using the ABI Prism Big Dye^®^ Terminator sequencing kit version 3.1 and run on the ABI 3130xl automated DNA sequencer (Applied Biosystems, Foster City, California, USA), according to manufactures instructions. Primers spanning the full-length integrase (867 bp) were used to sequence the PCR products in both directions. These included sequencing primers Poli6 and Poli7, and additional sequencing primers were used, namely Poli2 (TAAARACARYAGTACWAATGGCA), relative to position 4745–4766, and KLVO83 (GAATACTGCCATTTGTACTGCTG), corresponding to position 4750–4772.

### 2.5. Consensus Sequence Alignment and Mutation Detection

We performed a search on the HIV Los Alamos National Library (LANL) database (https://www.hiv.lanl.gov/components/sequence/HIVsearch.com). Our search inclusion criteria included all Cameroonian HIV-1 subtype CRF02_AG IN sequences identified from treatment-naïve patients. We selected one sequence per patient, and all problematic sequences were excluded from further analyses. The consensus sequence representing CRF02_AG was generated using the CRF02_AG study sequences (*n* = 37) as previously reported [25], accession number: MN816445-MN816488, while the consensus sequence for subtype C was derived from cohort sequences (*n* = 91,) as previously reported [26]. Nucleotide sequences were verified for stop codons, insertion, and deletions using an online quality control program on the HIVLANL database (https://www.hiv.lanl.gov/content/sequence/QC/index.htm). Multiple sequence alignments were done with MAFFT version 7, from which the consensus sequence was derived [27]. As part of quality control, each of the viral sequences were inferred on a phylogenetic tree in order to eliminate possible contamination. The amino acid sequence alignment was extensively screened for the presence of primary and secondary resistance-associated mutations (RAMs) and NOPs associated with resistance to known INSTIs.

### 2.6. Protein Modelling

A three-dimensional model was constructed for HIV-1C IN and recombinant form CRF02_AG using Schrodinger Prime modelling software [28,29]. A suitable homologous template was identified by performing a Blastp search using the consensus amino acid sequences of HIV1C IN and recombinant form CRF02_AG. Prior to modelling, missing residues were fixed by re-modelling the structure of template 5U1C using Schrodinger PRIME modelling software [30]. The Cryo-EM-solved IN subtype B intasome structure (ID: 5U1C) was used as the homologous template for comparative modelling, as it shared a high sequence identity and coverage with HIV-1C IN from a South African cohort and with the recombinant form CRF02_AG from a Cameroonian cohort. Additionally, as 5U1C contains a mutation at residue 152, this was mutated back into glutamic acid during re-modelling.

### 2.7. Protein Preparation

Processing of protein models was performed using Schrodinger Protein Preparation Wizard, which added hydrogen atoms, created disulphide bonds, assigned bond orders, filled in any missing side chains, and optimized the H-Bonds [31]. Magnesium ions were extracted from the prototype foamy virus and simian immunodeficiency virus IN experimental structures.

### 2.8. Model Validation

To assess the quality of the constructed IN protein models, a variety of structural parameters were tested within each model. The Structural Analysis and Verification Server (SAVES) was used for this purpose and includes the tools Procheck, Whatcheck, Prove, Verify3D, and ERRAT [32,33,34,35,36]. The cut-off values used by the tools were as follows: >80% for Verify3D, <1% for Prove, >50 for ERRAT. Whatcheck and Procheck are further subdivided into tests, but both make use of a Ramachrandran plot analysis, which is deemed passed if the majority of residues are within the allowable region. Furthermore, the root mean square deviation (RMSD) analysis was conducted using PYMOL/Maestro molecular visualizing software to compare backbone structural similarity to the experimentally solved 5U1C template structure [37].

### 2.9. INSTI Extractions and Molecular Docking

The INSTI’s RAL, EVG, DTG, and BIC were extracted from known solved structures deposited inside the Protein Data Bank (PDB) with the respective identifiers 3oya, 3l2u, 3s3m, and 6rwm [38,39,40,41]. The structures were superimposed upon our generated IN models using PYMOL and subsequently saved as protein–ligand complexes.

The INSTI CBT has not been published in complex with an IN protein, therefore molecular docking was done to predict the binding mode, affinity, and chemical interactions. Docking was performed using SMINA, a fork of AUTODOCK VINA [42,43]. The three-dimensional (3D) structures for CBT were acquired from the ZINC database [44]. Conversion of receptor and ligand structures from the respective pdb and sdf formats to pdbqt was done using OBABEL [45]. The docking grid was centered on the active site with a box size of 20Å in all planes. CBT was docked to each subtype structural intasome HIV-1C, B and AG, respectively. Ranking of generated binding poses was done based on the binding affinity values calculated using the Vinardo Scoring function with the top binding pose complex selected for the next step [46].

### 2.10. Refinement and Energy Minimisation

The refinement process involved repeating the aforementioned protein preparation steps. The complexes were subsequently energy-minimized using the CHARMM-GUI webserver [47] for structural preparation and energy minimization using default CHARMM-GUI-generated parameter files. The molecular dynamics engine software utilized was GROMACS 2018.1 [48] and charmm36M forcefield [49].

### 2.11. Binding Affinity Calculation and Interaction Analysis

The energy-minimized structures were desolvated and the ions removed using PYMOL. The complexes were separated into two separate files, one for IN as the receptor(s) and one for the INSTIs as the ligand(s). To calculate the binding affinity, the score only function was used within SMINA in combination with the Vinardo scoring function. Pymol was used to calculate polar interactions formed between the IN protein, DNA, MG, and the drugs.

### 2.12. Binding Site Analysis

The spatial and chemical features of the active sites containing the DDE motif of each IN subtype were compared to one another using PYMOL/MAESTRO. Briefly, the residues encompassing the binding site were determined by aligning each IN-INSTI complex to one another and extracting all residues within a 5Å radius of the inhibitor binding site. The binding sites were superimposed to determine the difference by measuring the Root Mean Square Deviation (RMSD) values for the backbone atoms of the protein chains.

## 3. Results

### 3.1. Sequence Alignments and Protein Structure Assessment

Two IN consensus protein sequences, corresponding to the South African and Cameroonian cohorts respectively, were aligned to the sequence of the subtype B IN structure (ID:5u1c) (Figure 1). The sequence identity was calculated to be 98% for both sequences when compared with subtype B. The alignment revealed 13 NOPs within the Cameroonian cohort-derived CRF02_AG IN sequences. The identified polymorphisms being K14R, V31I, V72I, L101I, T112V, T124A, T125A, K136T, I151V, V201I, T206S, V234I, S283G. The alignment also showed 17 NOPs within a South African cohort-derived Subtype C consensus sequence, namely D25E, V31I, M50I, V72I, F100Y, L101I, T112V, T124A, T125A, K136Q, I151V, V201I, T218I, V234I, R269K, D278A, S283G (Appendix A). The constructed IN models were validated with the following scores obtained with the SAVES server tests. CRF02_AG IN: Verify3D, 71%; ERRAT, 93/100; Prove, 7.8%; and for the Ramachandran plot analysis, 86.1% of residues are within the most favored region. Subtype C IN: Verify3D, 74%; ERRAT, 92/100; Prove, 7.8%; and for the Ramachandran plot analysis, 86.1% of residues are within the most favored region. Subtype B IN: Verify3D, 71%; ERRAT, 92/100; Prove, 10.3%; and for the Ramachandran plot analysis, 82.6% of residues are within most favored region.

The RMSD analysis score was found to be approximately 0.4Å when all three structures were compared to one another. In addition, there was minimal difference in secondary structural makeup (Figure 2). However, reference subtype B structure forms an extra helical turn absent in Subtype C and CRF02_AG.

### 3.2. Molecular Docking and Interaction Analysis

The molecular docking methodology was validated using the genetics algorithm (AutoDock), providing a correlation of the experimental data (half-maximal effective concentration, EC50) with the binding affinities showing reliable statistics (Appendix A, Figure 3) [50]. Subsequently, rescoring/docking assays was done using VINA of the five FDA approved and late-phase clinical trial INSTI’s showed minimal differences in binding affinity between one another, with less than 2 kcal/mol difference observed. The obtained binding affinities furthermore were comparable when SMINA was used to rescore Prototype Foamy Virus (PFV)-INSTI or Simian Immunodeficiency (SIV)-INSTI complexes. (Table 1). In Table 2, most of the drugs made two ionic interactions with Magnesium ions, except for EVG and BIC in HIV-1B IN due to the different orientation of the active site as a result of remodelling missing residues.

### 3.3. Binding Site Analysis

We calculated differences in total surface area, with 860Å^2^ for subtype B IN, 969Å^2^ for subtype C IN, and 1041Å^2^ for CRF_02_AG IN. RMSD analysis show that subtypes C IN and CRF_02_AG deviate by 0.309Å and 0.44Å, respectively. The NOP I151V was found to occur within the binding site, but does not directly interact with INSTI’s. NOP I151V is present in both subtype C and CRF_02_AG IN’s.

## 4. Discussion

The NOPs that have been identified in subtype C and CRF_02_AG IN have not been previously associated with HIV-1 IN drug resistance, except for the polymorphism M50I. M50I was identified in our subtype C IN sequences and this polymorphism has been reported to reduce DTG susceptibility when found in combination with the mutation R263K in HIV-1 subtype B IN [51,52]. However, in our study, the mutation R263K was not present. Furthermore, M50I is not able to cause drug resistance on its own but increases the effect of resistance exhibited by R263K [51]. In our study, M50I had no effect on INSTI’s binding to IN. NOPs were found to be occurring within their natural prevalence rates and these NOPs have minimal effect on INSTI susceptibility when occurring alone, such as in the case of M50I [51]. This is in contrast to the study by Brado et al. that reported an impact of NOPs on the stability of the protein complex, suggesting they may contribute to an overall potency against INSTIs. Our study is, however, in agreement with a study by Chitongo et al., 2020 that showed that one known major resistance mutation, G140S, had an effect on DTG drug binding in HIV-1C IN in combination with NOP’s. Therefore, NOPs alone have a negligible effect on drug binding [53].

A comparison of the backbone structures of the modelled IN proteins showed high structural similarity with one another with RMSD values less than 0.4Å, suggesting similar fold between the protein structures. Our IN homology modelling showed one slight secondary structural feature alteration within the N-terminal domain. Subtype B displayed a helical turn absent in Subtype C and CRF02_AG IN. Secondary structural features, such as helices, may influence the accessibility of drugs to a protein’s active site, either by directly changing the binding pocket properties or through affecting stability of the whole protein [54]. In our study, the structural alteration had no effect on drug binding as it is located further away from the active site.

Extractions showed that all INSTIs are able to bind to either of the tested IN subtype structures with plausible binding poses, which is in agreement to the previously reported PFV or SIV binding poses. No significantly reduced binding affinity was observed for each of the INSTIs, implying no negative alteration to the binding site which may prevent INSTI drug binding. Differences in binding affinity are present. The Binding affinity is an indication of how strong the ligand is binding to the active site. NOPs associated with a reduction to EVG susceptibility have been previously reported to occur with a relatively high frequency within CRF02_AG IN, however those NOPs were not identified in our study [55,56,57].

Magnesium ions are responsible and important for the binding of DNA. INSTIs competitively inhibit this process by binding to the magnesium ions [58,59,60]. It was therefore expected that interaction analysis would reveal that an interaction(s) takes place between the INSTIs and magnesium ions present within IN active sites. The interaction(s) with magnesium ions are considered to be essential for inhibition to take place, whereas other interactions are considered nonessential for the activity of INSTIs to take place. Therefore, we considered an INSTI to be successfully bound if interaction analysis predicted interactions occurring with magnesium ions. It may, however, be plausible that NOPs favor or reduce the likelihood of additional interactions occurring and thereby enhancing binding affinity. In our study, all drugs make contact with MG ions but only 1MG contact is found for HIV-1B IN bound to EVG and BIC. The remodeling of missing residues could be a reason for HIV-1B not making two MG ion contacts with EVG and BIC, but this is a surprising finding that warrants further investigation.

Binding site analysis further supports the results from the rescoring/molecular docking assay. The analyses conducted show that binding sites between each IN structure are identical. One NOP I151V is present within the binding sites of our study, but does not induce any effect beyond a slight spatial change due to its different side chain orientation.

The free energy of binding values calculated between the different proteins and drugs using the Molecular Mechanics Poisson-Boltzmann Surface (MM-PBSA) package confirmed no significant difference in the strength of binding between the drugs and different HIV-1 IN protein subtypes. However, the inhibition constant (Ki) values and inhibitor potency EC50 values provided evidence for DTG and BIC as strong binders with high inhibition values, suggesting DTG and BIC as good candidates for treatment of the three caused by the three different HIV-1 subtypes.

Future work should include viral integration assays to determine if DTG, BIC, and CBT can prevent viral integration within plasmids containing different HIV IN subtype sequences.

## 5. Conclusions

Our study showed that unique polymorphisms within geographically distinct HIV-1-infected populations with different variants do not prevent INSTI binding. However, linear correlation between the inhibition constant (Ki) values derived from docking experiments and experimental EC50 values for INSTI’s suggest DTG and BIC as strong inhibitors. We therefore put forward that second-generation DTG and BIC should be added to antiviral regimens as part of first-line regimens for HIV-1C IN subtype infections, this would account for cross-resistance which may occur between EVG and RAL as DTG and BIC may have a higher genetic barrier to resistance.

## Figures and Tables

**Figure 1 viruses-12-00936-f001:**
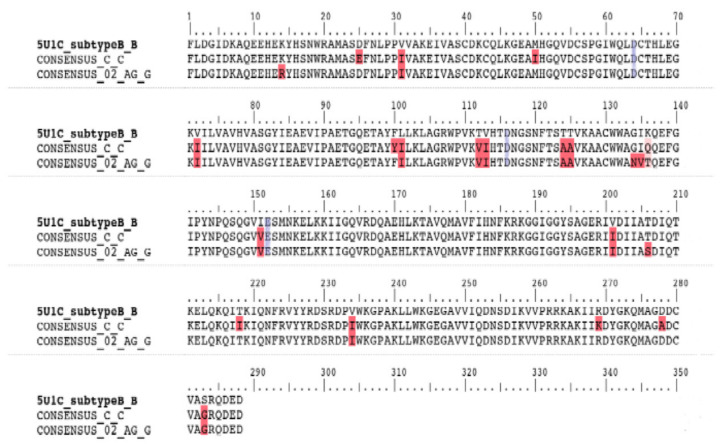
Amino acid sequence alignment of integrase variants subtype B, subtype Cza, and CRF_02_AG, respectively. Red highlighted residues indicate polymorphism locations in comparison with the Subtype B template. Blue highlighted residues indicate the aspartate (D64), aspartate (D116) and glutamate (E152) DDE motif.

**Figure 2 viruses-12-00936-f002:**
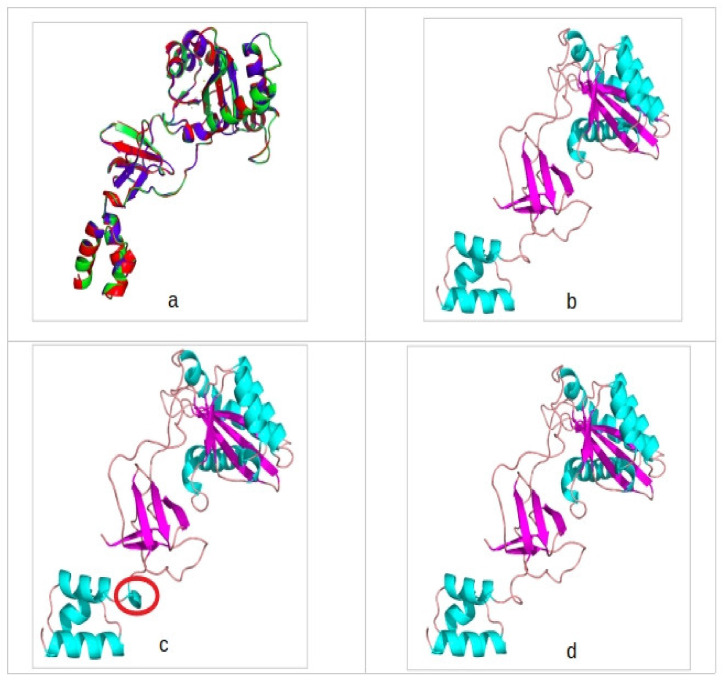
(**a**) A superimposition of all three integrase models showing high backbone identity and secondary structure conservation. (**b**) CRF02_AG integrase (IN) model colored according to secondary structure and in cartoon depiction. (**c**) Subtype B IN model colored according to secondary structure and in cartoon depiction. (**d**) Subtype C IN model colored according to secondary structure and in cartoon depiction. Encircled in red is the secondary structure difference observed between the template structure and the generated IN models. Light blue indicates the helices, purple indicates beta-sheets, and tint color indicates loops.

**Figure 3 viruses-12-00936-f003:**
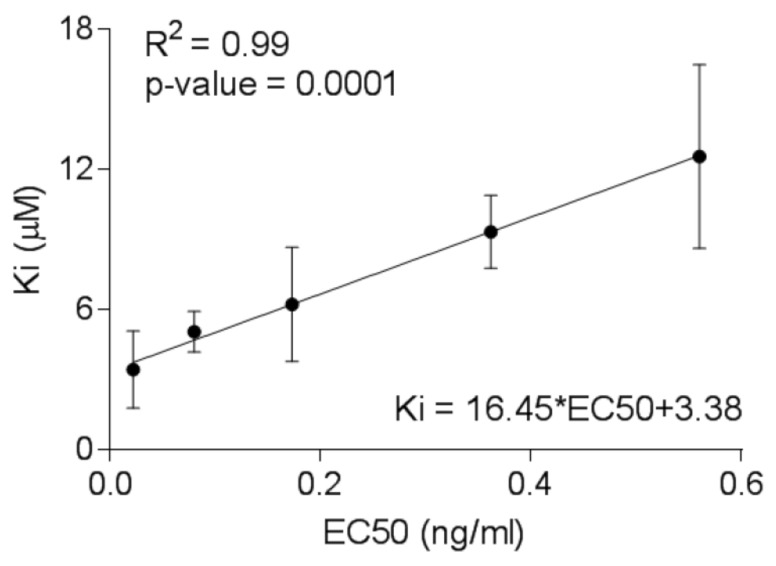
Linear relationship between the predicted binding constants calculated from the docking studies of integrase strand transfer inhibitors (INSTI) with HIV Ins and the experimental half-maximal effective concentration (EC50) values determined from the TZM-bl cells-based assay.

**Table 1 viruses-12-00936-t001:** Vinardo binding affinity scores predicted for each INSTI bound to the three IN subtypes CRF02_AG, B, and C and the PFV/SIV reported INs.

Drug	CRF02_AG (Kcal/Mol)	Subtype B (Kcal/Mol)	Subtype C (Kcal/Mol)	PFV/SIV (Kcal/Mol)
Raltegravir	−7.2	−4.4	−5.1	−6.1
Elvitegravir	−3.7	−3.4	−3.4	−4.0
Dolutegravir	−3.0	−3.4	−3.0	−5.4
Bictegravir	−4.0	−3.5	−3.5	−3.9
Cabotagravir	−6.5	−5.7	−7.0	N/A

**Table 2 viruses-12-00936-t002:** Summary of all interactions observed between the five INSTIs and three IN subtypes. Listed also are interactions which occur between INSTI and PFV/SIV nucleic acid.

INSTI	HIV-1B (ID:5u1c)		HIV-1C IN		CRF_02_AG IN		PFV/SIV IN	
	Hydrogen bonds	Ionic contact	Hydrogen bonds	Ionic contact	Hydrogen bonds	Ionic contact	Hydrogen bonds	Ionic contact
RAL	2 (CYS10, GLU152)	2MG	6 (THY11, GUA22, ASP64, ASP116, TYR143, GLU152)	2MG	5 (THY11, GUA22, ASP64, ASP116, GLU152)	2MG	4 (ASP128, ASP185, TYR212, GLU221)	2MG
DTG	5 (THY11, GUA22, ASP64, ASP116, GLU152)	2MG	4 (THY11, GUA22, ASP64, GLU152)	2MG	4 (ADE21, GUA22, ASP64, ASP116)	2MG	3 (ASP128, ASP185, GLU221)	2MG
EVG	5 (THY11, ADE21, GUA22, ASP116, ARG231)	1MG	3 (ADE21, GLU52, ASP64)	2MG	6 (THY11, GUA22, ASP64, ASP116, GLU152, ARG231)	2MG	1 (GLU221)	2MG
BIC	4 (THY11, GUA22, ASP64, CYS65)	1MG	4 (THY11, GUA22, ASP64, CYS65, ASP116)	2MG	5 (THY11, ASP64, CYS65, GLU92, ASP116)	2MG	3 (ASP64, ASP116, GLU152)	2MG
CBT	7 (THY11, ADE21, GUA22, ASP64, THR66, HIS67, ASP116)	2MG	8 (THY11, ADE21, GUA22, ASP64, THR66, HIS67, ASP116, GLU152)	2MG	6 (THY11, ADE21, GUA22, ASP64, HIS67, ASP116)	2MG	N/A	N/A

Number outside bracket indicates total number of interactions. Appendix A. RAL, raltegravir; DTG, dolutegravir; EVG, elvitegravir; BIC, bictegravir; CBT, cabotegravir.

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
