# Peer review of "Structural Comparison of Diverse HIV-1 Subtypes using Molecular Modelling and Docking Analyses of Integrase Inhibitors"

_viruses, 2020, doi:10.3390/v12090936_

Round 1

Reviewer 1 Report

Isaacs et al. have greatly improved the manuscript but there are a few issues that should still be addressed before I can recommend the manuscript for publication. Unfortunately, there are no general repositories for models built by homology or for compound docking predictions that I am aware of. This means that the value of the calculations from the models in the paper must be evaluated based on the figures provided by the authors. The supplemental figures are improved in the revised manuscript but are over-simplified still leave many details unanswered. For example, in supplemental figure S2 (d) what are the other interactions stabilizing the compound other than the two magnesiums? If these are the only stabilizing contacts, I doubt the compound would bind. It would also be helpful to color atom type for the compound and well as the protein in these figures. The reason I feel these details are important is that calculations such as binding affinities is problematic even from empirically determined structures due to protein conformational heterogeneity. In cases where the binding site is rigid these calculations may be more valid, but less so if the conformation changes. Presenting these details for the reader allows the reader to come to his/her own conclusions regarding the validity of the calculations from the model. An alternative would be to make the final model coordinates for each complex available upon request with a statement to such effect in the text.

Reviewer 2 Report

The manuscript was greatly improved in revision. However, the models obviously need more refinement. Most troubling is Table 2, which seems to suggest that some compounds do not make interactions with the 2 Mg ions in some cases. This goes against a lot of prior data (including a large volume of high resolution structural data) suggesting that the INSTIs are two-metal chelators! I am not saying this is impossible, but if it is the case, the models where INSTIs bind the active site without making contacts to both Mg ions need more justification, description and experimental validation. Although most likely, there was something wrong with restraints during energy minimization or with initial model assembly. Or perhaps there is a problem with the data reported in this table.

Generally, classical MD does not handle well metal interactions, but it can be done. Given the apparent loss of metal binding, it is a mystery how the authors were able to calculate similar binding affinities (Table 1)!  

A minor issue:

Images in Fig2 seem to be stretched vertically (?).

Table 2, again: 5U1C (the first column) is a strand transfer complex, it does not have any INSTIs bound. Moreover, the construct had a mutation inactivating the enzyme (E152Q, presumably to block disintegration activity).

Round 2

Reviewer 1 Report

Paper is acceptable in present form